# Discovery of Sulfated Small Molecule Inhibitors of Matrix Metalloproteinase-8

**DOI:** 10.3390/biom10081166

**Published:** 2020-08-09

**Authors:** Shravan Morla, Umesh R. Desai

**Affiliations:** 1Department of Medicinal Chemistry, Virginia Commonwealth University, Richmond, VA 23298, USA; smorla@scripps.edu; 2Drug Discovery and Development, Institute for Structural Biology, Virginia Commonwealth University, Richmond 23219, VA, USA

**Keywords:** matrix metalloproteinases, glycosaminoglycans, small molecule inhibitors, drug discovery

## Abstract

Elevated matrix metalloproteinase-8 (MMP-8) activity contributes to the etiology of many diseases, including atherosclerosis, pulmonary fibrosis, and sepsis. Yet, very few small molecule inhibitors of MMP-8 have been identified. We reasoned that the synthetic non-sugar mimetics of glycosaminoglycans may inhibit MMP-8 because natural glycosaminoglycans are known to modulate the functions of various MMPs. The screening a library of 58 synthetic, sulfated mimetics consisting of a dozen scaffolds led to the identification of only two scaffolds, including sulfated benzofurans and sulfated quinazolinones, as promising inhibitors of MMP-8. Interestingly, the sulfated quinazolinones displayed full antagonism of MMP-8 and sulfated benzofuran appeared to show partial antagonism. Of the two, sulfated quinazolinones exhibited a >10-fold selectivity for MMP-8 over MMP-9, a closely related metalloproteinase. Molecular modeling suggested the plausible occupancy of the S_1_^′^ pocket on MMP-8 as the distinguishing feature of the interaction. Overall, this work provides the first proof that the sulfated mimetics of glycosaminoglycans could lead to potent, selective, and catalytic activity-tunable, small molecular inhibitors of MMP-8.

## 1. Introduction

Matrix metalloproteinase-8 (MMP-8), also known as neutrophil collagenase or collagenase-2, is a Zn^+2^-dependent endopeptidase of the MMP family. It is primarily expressed and secreted by neutrophils as a zymogen (pro-MMP-8), and is activated by the reactive oxygen species released from activated neutrophils, thus rendering MMP-8 as having a key role in both acute and chronic inflammation [1]. Tissue inhibitors of metalloproteinases (TIMPs) regulate the activity of MMP-8 by forming a 1:1 stoichiometric inhibitory complex [2]. However, imbalances in MMP-8 and TIMP levels can lead to increased collagen degradation and pathological remodeling of the extracellular matrix [1]. Although it was initially believed that MMP-8′s only function was to degrade collagen, later studies have identified both non-collagenous and non-structural substrates [1], thereby implicating MMP-8 in various pathological processes (see Table 1 for the full listing). In recent years, a wealth of preclinical and clinical data have emerged that correlate high MMP-8 levels with the potential for diagnosing diseases [3]. Likewise, MMP-8 inhibitors have also been suggested as potential therapeutics for treating some of these pathologies [3]. 

Most early attempts to develop MMP inhibitors focused on developing peptide-based agents mimicking endogenous substrate sequences aided by a hydroxamate group, which ligates the critical Zn^+2^ co-factor [3,16]. Unfortunately, the high affinity of hydroxamates to Zn^+2^ containing proteins, which include several MMPs, as well as a disintegrin and metalloproteinase (ADAM) and ADAM with thrombospondin motifs (ADAMTS) family members, introduced a significant non-selectivity of inhibition, causing severe side-effects in the clinical trials [3,16]. Later, researchers focused on the development of peptides with a lower affinity to Zn^+2^ by including phosphonates, thiolates, and carboxylates in the structure (see Appendix A) [3,16,17,18]. Although such peptides demonstrated significantly reduced side-effects, they failed to show sufficient efficacy in the clinical trials [3]. This resulted in a rather negative perception of MMPs as drug targets in the community [3].

After nearly a decade of no progress in clinical trials with MMP inhibitors, a better understanding of the MMP biology has led to the insight that the initial clinical trials were perhaps held prematurely [3,16]. Learning from both the fundamental advances made so far and the limitations of the broad-spectrum non-selective MMP inhibitors, researchers have focused on the development of selective small molecule MMP inhibitors. This has led to the development of selective MMP-2, -9, and -13 inhibitors. Several of these molecules are now being tested in clinical trials [3,16]. 

In contrast to these successes, MMP-8 appears to be a rather challenging target. Although small molecule inhibitors are likely to prevent and treat multiple diseases (Table 1), few reports are available on the development of MMP-8 inhibitors. In fact, only two publications have reported MMP-8 inhibitors in the last decade [19,20]. Thus, the vast chemical space offered by small molecules appears to be almost untouched for MMP-8 inhibitor development.

Natural glycosaminoglycans (GAGs) modulate various functions of MMPs, including cell localization, conformation, and stability [21,22]. As one would expect, based on the extensive literature behind GAG–protein interactions [23,24], GAGs are expected to affect MMP activity, depending on the length of their polymeric chains and the structure of the local sulfated microdomains [21,25]. In fact, distinct GAGs have been found to inhibit MMP-9, which is closely related to MMP-8 [26]. Furthermore, the potency of natural GAGs against MMPs is quite promising. However, the heterogeneity of these GAGs is problematic for drug development. Additionally, polymeric natural GAGs generally display unwanted side-effects; they are impossible to chemically synthesize, and their bio-analytical characterization is a major challenge. These challenges reduce the enthusiasm to pursue natural GAGs as drugs [27]. 

In contrast, sulfated small molecules that mimic GAGs offer much better prospects. These molecules, referred to as non-saccharide GAG mimetics (NSGMs), have now been shown to bind and functionally modulate multiple GAG-binding proteins [28,29,30,31,32,33,34]. For example, distinct NSGMs have been identified as ligands of fibroblast growth factor receptor-1 [28], antithrombin [33,34], neutrophil elastase [29], coagulation factor Xia [30,35], plasmin [31], and viral glycoprotein D [32], each of which are known to bind to heparin. In fact, several NSGMs have been found to display one-to-one correspondence with specific GAG sequences [28,36]. In terms of drug-like properties, NSGMs are fully synthetic, which bodes well for analytical ease and scalability of preparation [27,37]. Their much smaller molecular size (MW 500–2000) in comparison with natural GAGs (MW 15,000–100,000) enables a rational and/or computational design for identifying advanced analogs [28]. Additionally, NSGMs are highly water-soluble owing to the presence of one or more sulfate groups, which enables easier formulation, e.g., direct instillation into the lung. This property could have important value for lung disorders, such as chronic obstructive pulmonary disease (COPD) and idiopathic pulmonary fibrosis (IPF), where MMP-8 inhibition has shown to be beneficial.

Over the past decade, we have developed a library of NSGMs (Figure 1) with 12 different chemical scaffolds (apigenin, benzofuran, catechin, glucoside, gossypetin, inositol, luteolin, morin, phloretin, quercetin, quinazolinone, and resveratrol). The library contains mimetics with different chain lengths (monomer, dimer, and trimer), as well as varying sulfation levels (mono-sulfate through dodeca-sulfate), which affords an excellent potential to mimic the diversity of the chain length and sulfation patterns of GAGs. 

The rationale behind the development of the NSGM library was that while the sulfate groups will offer GAG-like protein recognition features, the aromatic scaffold will impart hydrophobic characteristics. These dual ionic and non-ionic forces of recognition are likely to enhance selectivity for the target protein of interest. Using this concept, distinct NSGMs have been developed as promising anticoagulants [35], antifibrinolytics [31], anticancer [39], anti-inflammatory [29], and antiviral agents [32]. More importantly, our work has shown that initial “hits” can be systematically developed through an appropriate hit-to-lead optimization strategy into potent and selective candidates, with favorable pharmacokinetic properties and minimum adverse side-effects [30,36,40,42]. 

In this work, we screen a library of NSGMs using computational and in vitro techniques to identify promising inhibitors of MMP-8. Our work shows, for the first time, that MMP-8 can be inhibited by targeting electropositive GAG binding sites, and offers at least one promising small molecule inhibitor of MMP-8.

## 2. Materials and Methods

### 2.1. Materials

Human pro-MMP-8 was purchased from R&D Systems (Minneapolis, MN, USA). p-aminomercuric acetate and MMP-8 fluorogenic substrate (DNP-Pro-Leu-Ala-Tyr-Trp-Ala-Arg) were obtained from Sigma-Aldrich (St. Louis, MO). MMP-9 inhibitor screening assay kit was obtained from Abcam (Cambridge, UK). All other reagents were purchased from Fisher Scientific (Waltham, MA, USA).

### 2.2. Chemistry

The synthetic scheme of the NSGMs was previously reported [30,31,38,39,40,41]. All the molecules were characterized by NMR and ultra-performance liquid chromatography (UPLC) − electrospray ionization mass spectrometry (ESI-MS). The purity of each agent was >95%, as analyzed by UPLC-MS.

### 2.3. Direct Inhibition Studies

NSGMs were screened for the inhibition of MMP-8 using a fluorogenic substrate assay, as described earlier [20]. Briefly, 100 µg/mL pro-MMP-8 was activated by incubating with 1mM p-aminomercuric acetate at 37 °C for 1 hr. Activated MMP-8 (10 nM, final) was then incubated with NSGMs (100 µM, final) in Tris-buffered saline containing 10 mM CaCl_2_,1 µM ZnCl_2_, pH 7.5 for 10 min at 37 °C. Residual MMP-8 activity was measured by adding the fluorogenic substrate (DNP-Pro-Leu-Ala- Tyr-Trp-Ala-Arg, 20 µM, final) and monitoring initial linear rate of increase in fluorescence. IC50s were calculated using the following equation, $Y=Y_{0}+\frac{Y_{M}-Y_{0}}{1+10^{\left(log{\left[I\right]}_{0}-logIC_{50}\right)\times HS}}$. Y is the ratio of residual enzyme activity in the presence of NSGM to that in its absence, Y_0_ and Y_M_ are the minimal and maximal values of Y, respectively, obtained following regression, and HS is the Hill slope of inhibition. 

Screening of sulfated benzofurans and quinazolinones (100 µM, final) against MMP-9 was performed using a commercially available assay kit (Abcam catalog #ab139448) following manufacturer’s protocol. All experiments were performed at least twice in duplicates. 

### 2.4. Molecular Modeling Studies

The X-ray crystal structures of MMP-2 (PDB ID: 1CK7), MMP-8 (PDB ID: 5H8X) and MMP-9 (PDB ID: 2OW1) were retrieved from RCSB protein data bank and the electrostatic surface potential (ESP) maps calculated using PyMOL (Molecular Graphics System, Schrödinger, NY). MMP-8 was prepared for docking by removing water molecules, adding polar hydrogens, Kollman charges and solvation parameters using MGL (Molecular Graphics Laboratory) Tools of AutoDock 4.2 (Scripps Research, CA) to generate pdbqt files. Docking was performed in two stages. First, to identify a potential binding site of NSGMs, a blind docking was performed using a grid box of 100 × 100 × 100 Å with 0.375 Å grid spacing to include all the amino acid residues. A short search of 10 runs with 250,000 evaluations per run was performed with Lamarckian Genetic Algorithm (LGA) docking. Later, based on the results of this initial blind docking, a more stringent grid with a radius of 16 Å was defined and the evaluations increased to 25000000. The top 10 binding poses were then individually analyzed, and the pose favoring the best score with least root-mean-square difference was chosen. The interactions of the docked poses were analyzed using PyMOL or LigPlot+ (European Molecular Biology Laboratory, Heidelberg, Germany).

## 3. Results and Discussions

### 3.1. GAG Binding Potential of MMP-8

To assess whether MMP-8 is likely to interact with GAGs, we evaluated its electrostatic surface potential (ESP), as discussed in the literature (Figure 2A) [43]. The ESP map of MMP-8 revealed a catalytic site surrounded by electropositive residues that may serve as GAG or NSGM interacting residues. In fact, multiple electropositive microdomains are located close by, which could possibly favor binding to dimeric and/or trimeric GAG-like NSGM molecules. Thus, we reasoned that there is a high probability of identifying an NSGM that inhibits MMP-8.

Interestingly, we noted that the S_1_^′^ pocket of MMP-8 is fairly electropositive in nature. The S_1_^′^ pocket is the most varied among the different MMPs. In fact, MMPs can be grouped in the order of the depth of their S_1_^′^ pockets. MMPs have either “shallow”, “intermediate”, or “deep” S_1_^′^ pockets [3,44]. MMP-8 is categorized as one with an “intermediate” S_1_^′^ pocket. Apart from the size of the S_1_^′^ pocket, the residues in the S_1_^′^ specificity loop are also considerably different among different MMPs [45]. These differences in the S_1_^′^ pocket and its specificity loop are currently being exploited to develop selective MMP inhibitors [44]. Thus, to assess if the electropositive nature of the S_1_^′^ pocket is common among MMPs in the “intermediate” category, we analyzed the ESPs of MMP-2 and -9, two other “intermediate” S_1_^′^ MMPs [3]. Surprisingly, both MMP-2 and -9 displayed an electronegative S_1_^′^ pocket in comparison with that of MMP-8 (Figure 2B,C). Considering these differences in the S_1_^′^ pocket of MMP-8 with closely related MMPs, we predicted that the inhibition of MMP-8 by GAGs or NSGMs that target the S_1_^′^ pocket is likely to yield a good selectivity.

### 3.2. Structure−Activity Relationships for the Library of NSGMs

To test our hypotheses, we screened our NSGM library against MMP-8 using a fluorogenic substrate hydrolysis assay, as described previously [20]. The screening of the NSGM library resulted in a wide range of inhibitor efficacies (0–100%) against MMP-8 (Figure 3). Interestingly, the inhibition appeared to be related to certain sulfated scaffolds with a majority, e.g., apigenins, catechin, glucoside, inositol, luteolin, morin, phloretin, quercetins, and resveratrol, showing minimal inhibition of MMP-8. This is unusual, because one would expect that the highly sulfated agents, e.g., those carrying more than six sulfate groups, could be expected to target electropositive surfaces more easily. Yet, the results imply that the electropositive regions on MMP-8 are very discriminatory.

Only five NSGMs (**26**, **38**, **40**, **41**, and **42**) demonstrated an excellent MMP-8 inhibition (>80%). These NSGMs belong to the sulfated benzofuran and sulfated quinazolinone scaffolds. When profiled for the quantitative measurement of MMP-8 potency (Figure 4), the five MMP-8 inhibitors were found to exhibit a three-fold range of potency (*IC*_50_ = 11–34 µM, Table 2).

At the level of the MMP-8–GAG system, these results are interesting on several fronts. First, the inhibition profiles support the hypothesis that the family of NSGMs is likely to yield an MMP-8 inhibitor, given its structural diversity. The five NSGMs could be classified as “hits” with moderate potency, which will require secondary drug design efforts to yield “lead(s)”. Second, the hit yield, i.e., only 5 out of the 58 studied, is relatively low, which suggests excellent weeding-out by MMP-8. Third, not all NSGMs inhibit MMP-8 fully (ΔY > 90%). The lone sulfated benzofuran **26** displays a partial inhibition profile (ΔY = 75%), which presents the possibility of regulating the MMP-8 activity (ΔY = 20–80%), rather than knocking it out completely. Recently, small molecule regulation of soluble enzymes has been discovered and was found to offer beneficial properties [38,46,47]. Similarly, regulation of the MMP-8–NSGM **26** system may offer the benefit of retaining basal levels of the MMP-8 activity that is critical for optimal growth. Fourth, the result that none of the highly sulfated mimetics (NSGMs **1**–**14**, **57**, or **58**), which carry more than six sulfate groups, were active against MMP-8 (Figure 3) implies significant contributions of the aromatic groups. This supports the hypothesis on dual ionic and non-ionic forces governing recognition.

Among the sulfated benzofurans, the closely related monomers (**15**–**25**) and dimers (**27**–**35**) did not inhibit MMP-8 in contrast to **26**, which was one of the three most active NSGMs. In each case, the structural changes were primarily in the terminal substituents (R_1_ and R_4_, Figure 1), suggesting a stringent size dependence. Among the sulfated quinazolinones **36**–**42**, agents with linkers of less than six carbons did not inhibit MMP-8. This implies that the NSGM binding site on MMP-8 is likely to contain two sub-sites spaced several angstroms apart.

### 3.3. Computational Analysis of the Preferred Site of NSGMs Binding to MMP-8

To identify a plausible site of binding of the active sulfated benzofurans and sulfated quinazolinones, we performed genetic algorithm-based molecular docking and scoring studies. Docking was performed using the available MMP-8 crystal structure (PDB ID: 5H8X) and AutoDock 4.2 (Scripps Research). The studies revealed that the sulfate group of NSGM **26** interacts with the catalytic Zn^2+^ and the triad of histidine residues (H197, H201, and H207, Figure 5A,B). This is similar to the previously reported interactions of MMPs with the phosphate groups [48]. Additionally, the oxygen atoms of the furan ring and the linker form a bidentate interaction with S151 (Figure 5B), and the aromatic substituent at the R_4_ position contributes towards hydrophobic and cation-Π interactions with residues in the unprimed S_2_ and S_3_ pockets. In contrast, other sulfated benzofurans did not have any interaction with S151, either because of their sub-optimal chain length (NSGMs **15**–**25**) or the bulk of their substituents (NSGMs **27**–**35**). This explained the observed lack of MMP-8 inhibition with these NSGMs. 

In the manner of NSGM **26**, sulfated quinazolinone dimers **36**–**42** also bound near the catalytic site of MMP-8 (Figure 5A,C). For NSGM **38**, one monomeric unit of the quinazolinone was found to insert deep into the S_1_^′^ pocket (shown as a pale-yellow surface in Figure 5A), which resulted in a strong binding of the sulfate with L193, L214, A220, and R222 (Figure 5C). The insertion of the quinazolinone ring into the S_1_^′^ pocket positioned the triazole linker close to the catalytic Zn^+2^ ion. However, based on these experiments, NSGM **38** did not make polar interactions with the catalytic Zn^+2^ or the histidine triad. Instead, the nitrogen atom of the triazole ring, along with the nitrogen of the quinazolinone ring form hydrogen bonds with the backbone oxygen of P217. Additional polar, hydrophobic, and cation-Π interactions were observed with multiple residues present in the site of binding (see Appendix A). The insertion of sulfated quinazolinone dimers (**36**–**42**) in the S_1_^′^ pocket of MMP-8 and their interactions with the residues in the S_1_^′^ specificity loop is particularly interesting (Figure 5A). Such GAG-like molecules inhibiting MMP-8 by binding in the electropositive S_1_^′^ pocket have the potential to be selective towards MMP-8. Thus, the identification of NSGM hits in this work is likely to be of significant value for both drug discovery and chemical biology campaigns against MMP-8.

Several other aspects of structure–activity dependence were also observed. For example, when one of the molecules of quinazolinone in the dimers that inhibited MMP-8 (**38**, **40**, **41**, and **42**) was replaced with quercetin, the resulting quercetin–quinazolinone heterodimers (**43**–**46**) lost their inhibition towards MMP-8 (Figure 3). Although the core scaffold of quinazolinones and quercetins studied here are very similar, i.e., fused bicyclic ring attached to a phenyl group, the presence of three sulfate groups on the quercetins compared with one sulfate on the quinazolinones adds both steric bulk and negative charges. Based on computational modeling, this addition of steric bulk drives the heterodimeric molecules away from the S_1_^′^ pocket (Appendix A), resulting in a loss of MMP-8 inhibition. 

### 3.4. Sulfated Quinazolinones Do Not Inhibit MMP-9

We next turned our attention to MMP-targeting selectivity studies. Considering that MMP-9 is the most closely related metalloenzyme to MMP-8, and as it belongs to the “intermediate” S_1_^′^ pocket category like MMP-8, we screened sulfated benzofurans and sulfated quinazolinones against MMP-9. NSGM **26** also inhibited MMP-9 reasonably well (Figure 6). This was not too unusual, because its inhibition of MMP-8 was predicted to arise from coordination with the catalytic Zn^2+^ and histidine triad (see Figure 5B), which is known to be the origin of a lack of selectivity associated with many inhibitors of MMPs. Another sulfated benzofuran, NSGM **29**, which did not inhibit MMP-8, also showed >80% inhibition of MMP-9. Thus, the future use of sulfated benzofurans as MMP-8 hits is likely to be beset with such non-selectivity concerns.

In contrast, the sulfated quinazolinones did not inhibit MMP-9 significantly (Figure 6), thus displaying >10-fold selectivity for MMP-8 over MMP-9. Considering that the predicted mode of binding of sulfated quinazolinones does not involve a strong engagement of the catalytic Zn^2+^ or the histidine triad (see Figure 5C), the observation of a lack of MMP-9 activity is encouraging. The sulfate group of **38**, inserted in the S_1_^′^ pocket, forms hydrogen and ionic bond interactions with A220 and R222, respectively, of MMP-8. At the corresponding positions in MMP-9, there is neither an arginine nor an alanine, which supports the observed lack of inhibition. In fact, the arginine at 222 is found in only one other MMP (MMP-28), and the alanine at 220 is not present in any other MMPs [45]. Thus, it is possible that NSGM **38** is an excellent hit with a high potential for selectivity of MMP-8. This hypothesis will need to be further experimentally validated through extended selectivity studies.

Overall, our “hit” NSGMs are the first group of GAG-related inhibitors of MMP-8. Given the urgent need and lack of efforts directed towards developing MMP-8 inhibitors, these NSGMs offer a unique starting point for further structure-guided hit-to-lead optimization studies. Two major optimization routes could be considered, namely: (1) Our modeling insights convey that the use of a biphenyl substituent, instead of a phenyl substituent, on the sulfated quinazolinone would lead to a greater reach of a potential inhibitor inside the S_1_^′^ pocket. This should enhance both the potency and selectivity. In fact, a peptide-hydroxamate containing a biphenyl substituent was previously shown to inhibit MMP-8 with a low nanomolar potency by occupying the S_1_^′^ pocket [49]. However, because of the strong Zn^+2^ chelating nature, metabolic instability, and toxicity associated with hydroxamates, these molecules failed in the clinical trials [3,16]. (2) Increasing the polarity of the linker by changing it from an alkyl-based chain to an ethylene glycol-based chain would likely result in polar interactions with the residues lining the S_2_ and S_3_ pockets. This should also enhance the potency and perhaps selectivity.

## 4. Conclusions

The present study shows, for the first time, the possibility of inhibiting MMP-8 by targeting electropositive GAG binding sites, and identifies two new small molecule GAG-mimicking sulfated scaffolds, the sulfated benzofurans and sulfated quinazolinones, as inhibitors of MMP-8. Of these, the sulfated benzofurans appear to be broad-spectrum MMP inhibitors because of their engagement of the catalytic Zn^2+^ and histidines. In contrast, the sulfated quinazolinones bind in the S_1_^′^ pocket of MMP-8, thereby enhancing the selectivity of inhibition. Considering that several NSGM-based initial hits have been transformed through hit-to-lead optimization strategies into potent and selective candidates [30,36], the potential to transform NSGM **38** into a promising drug-like candidate or a chemical biology tool is high. 

## Figures and Tables

**Figure 1 biomolecules-10-01166-f001:**
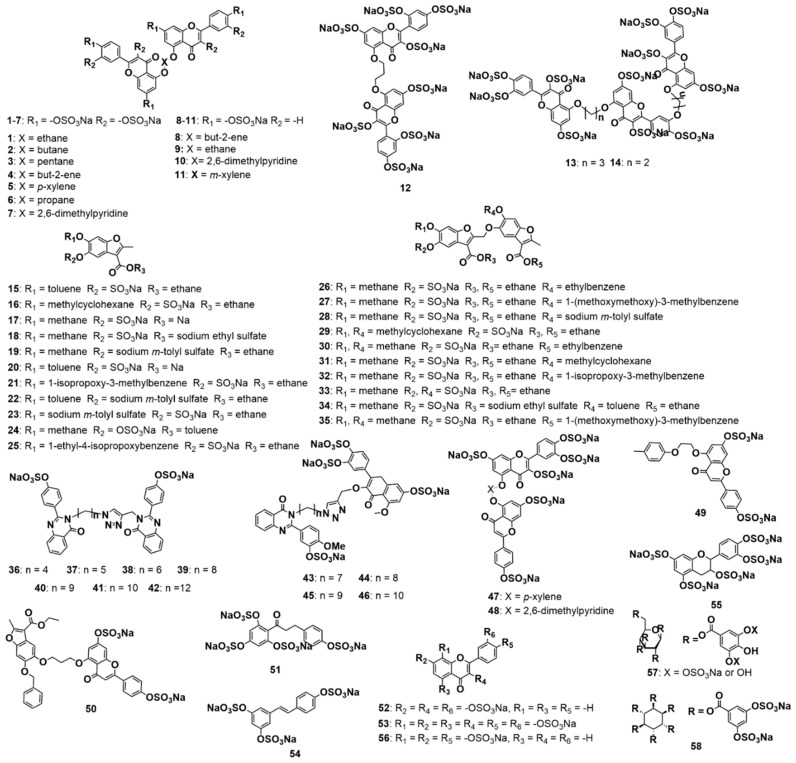
Structures of the 58 non-saccharide glycosaminoglycans (GAGs) mimetics (NSGMs) studied against MMP-8. The synthesis and characterization of these molecules has been previously reported [30,31,38,39,40,41].

**Figure 2 biomolecules-10-01166-f002:**
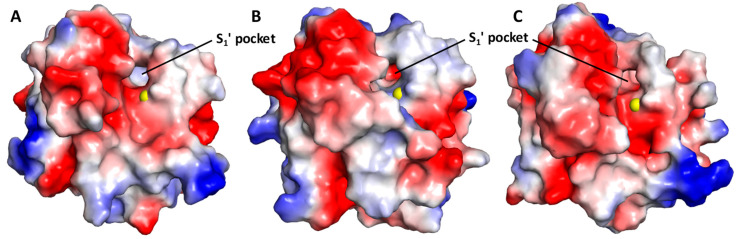
Electrostatic surface potential (ESP) map of MMP-8 (**A**), MMP-9 (**B**), and MMP-2 (**C**) showing electropositive regions (colored blue) that may serve as the site of binding for GAGs and NSGMs. Catalytic Zn^2+^ is shown as a yellow sphere. Note the differences in the nature of ESPs between closely related MMPs, particularly inside the S_1_^′^ pocket (blue vs. red), despite their relatively good sequence similarity/homology.

**Figure 3 biomolecules-10-01166-f003:**
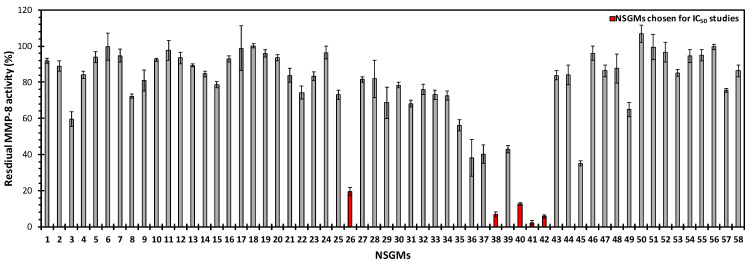
Residual activity of MMP-8 (10 nM) in the presence of each NSGM (100 µM) in Tris-buffered saline containing 10 mM CaCl_2_, 1 µM ZnCl_2_, pH 7.5 for 10 min at 37 °C. The residual MMP-8 activity was measured using a fluorogenic substrate. All of the measurements were performed at least in duplicate. Error bars represent ± 1 standard deviation (SD).

**Figure 4 biomolecules-10-01166-f004:**
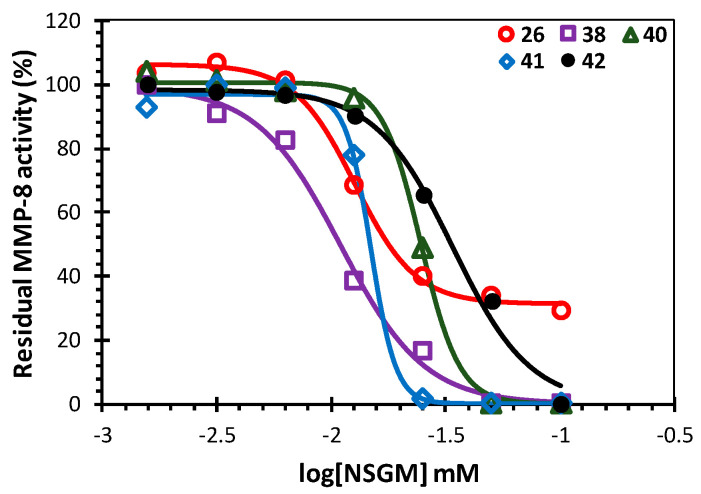
Direct MMP-8 inhibition profiles of the most promising NSGMs, including **26**, **38**, **40**, **41**, and **42**, at pH 7.5 and 37 °C. Solid lines represent the data fitted to the standard sigmoidal dose−response equation to derive IC_50_ and ΔY, which refer to the potency and efficacy of inhibition, respectively.

**Figure 5 biomolecules-10-01166-f005:**
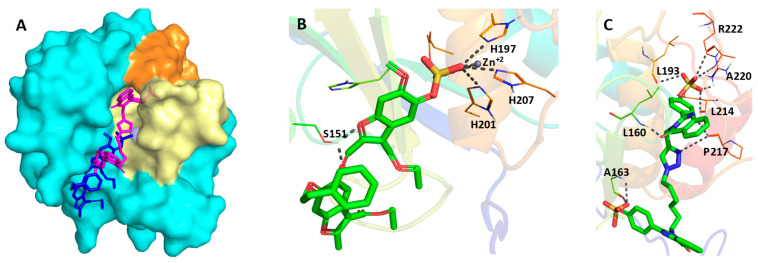
(**A**) Surface representation of MMP-8 (cyan) showing the potential interacting sites of sulfated benzofuran **26** (blue) and sulfated quinazolinone **38** (magenta). The S_1_^′^ pocket and specificity loop are shown in pale yellow and orange, respectively. (**B**,**C**) Polar interactions of NSGM **26** (**B**, shown in sticks) and NSGM **38** (**C**, shown in sticks) with MMP-8 (cartoon). Interacting residues of MMP-8 are shown as lines, catalytic Zn^+2^ ion as a sphere, and polar interactions as grey dotted lines.

**Figure 6 biomolecules-10-01166-f006:**
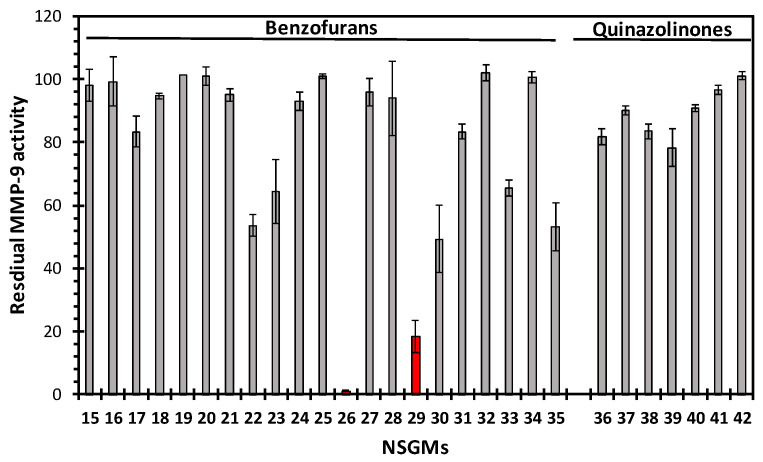
Screening of NSGM **15**–**42** at 100 µM against MMP-9. Experiments were performed using a commercially available assay kit (Abcam, Cambridge, U.K.) in at least duplicates. Error bars represent ± 1 SD.

**Table 1 biomolecules-10-01166-t001:** Matrix metalloproteinase-8 (MMP-8) in various diseases.

Disease	Role of MMP-8
Atherosclerosis	High MMP-8 in atheromatous and fibrous plaques promote plaque rupture, leading to vascular and cardiac events, including myocardial infarction, ischemic stroke, and abdominal aortic aneurysm [4]. MMP-8 knockdown significantly reduces atherosclerotic events in mouse models [5].
Bacterial meningitis	High MMP-8 levels in the cerebrospinal fluid of children with bacterial meningitis are associated with blood–brain barrier damage and neuronal injury [6].
Cancer	MMP-8 displays apparently contradictory roles in both cancer progression and inhibition, depending on the type of cancer, making it both a target and an anti-target for cancer therapy [7].
Chronic obstructive pulmonary disease (COPD)/emphysema	Increased MMP-8 levels lead to poor pulmonary function and emphysema severity [8].
Coronary artery disease	The plasma MMP-8 levels in patients with coronary artery disease is associated with disease severity [9].
Idiopathic pulmonary fibrosis (IPF)	MMP-8 levels in plasma, bronchoalveolar lavage fluid, and lung macrophages of IPF patients are noted to be high. MMP-8 knockdown protects mice from bleomycin-mediated lung fibrosis [10].
Obesity	MMP-8, which degrades the human insulin receptor, is increased in the serum of obese individuals, and may contribute to insulin resistance. MMP-8 inhibition restores the insulin receptor [11].
Periodontal diseases	Upregulated MMP-8 levels are observed in gingival cervicular fluid, corresponding to 90–95% of all collagenolytic activity. MMP-8 inhibitors cease the progression of periodontitis [12].
Sepsis	The increased gene expression and activity of MMP-8 correlates with disease severity and a worsening clinical outcome [13].
Tuberculosis	MMP-8 dependent tissue destruction is observed in patient lung biopsies [14].
Wound healing	Increased MMP-8 levels in mice prevent tissue repair, leading to impaired wound healing [15].

**Table 2 biomolecules-10-01166-t002:** Direct inhibition of MMP-8 by selected NSGMs.

NSGM	IC_50_ (µM)*^a^*	ΔY (%)*^a^*
**26**	13 ± 1*^b^*	75 ± 3
**38**	11 ± 1	99 ± 6
**40**	25 ± 1	100 ± 2
**41**	15 ± 1	97 ± 2
**42**	34 ± 4	98 ± 8

*^a^* Obtained by regression analysis of the dose-dependence of the MMP-8 activity in Tris-buffered saline containing 10 mM CaCl_2_, 1 µM ZnCl_2_, pH 7.5, at 37 °C. IC_50_ and ΔY refer to potency and efficacy of inhibition, respectively. *^b^* Error refers to ± 1 SD.

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
