# Peer review of "Discovery of Sulfated Small Molecule Inhibitors of Matrix Metalloproteinase-8"

_biomolecules, 2020, doi:10.3390/biom10081166_

Round 1

Reviewer 1 Report

The main idea of this short communication is innovative and describes the first proof that sulfated mimetics of glycosaminoglycans (GAGs) could lead to potent small molecular inhibitors of MMP-8. Mounting evidence have been shown about the key role of MMPs in many disorders and the discovery of selective inhibitors is an emergent research field since the side-effects of non-selective inhibitors was observed in the clinical trials. In addition, the present study is interesting because, while the present results were not obtained in vivo, the authors mentioned the non-saccharide GAG mimetics seems to have favorable pharmacokinetic properties and minimum adverse side-effects.

            The communication is well elaborated but is not clear why the authors reasoned that non-sugar mimetics of GAGs may inhibit MMP-8 as described in introduction. Thus, the introduction should be improved. The authors should emphasize why non-sugar mimetics GAGs were chosen to evaluate the MMP-8 inhibition.  The applied methods are appropriate to communication and well described. The results/discussion section is well described.

Specific Comments to be addressed by the Authors:

  1. The references should be checked again because they are not always properly quoted. For example, reference 2, is a review article about inhibitors and should not be mentioned referring the importance of MMP-8 in acute and chronic inflammation (line 35).
  2. The last column of the Table 1 (reference) needs to be formatted. Is this column necessary? Is the table necessary?
  3. The methods should be more detailed for the direct inhibition studies. For example, how many times the experiment was performed? The independent experiments was performed?
  4. Why the effects of non-saccharide GAG mimetics were not evaluated for the MMP-2, which is also closely related MMP-8?
  5. NSGM 26 inhibits reasonably well MMP-8 and MMP-9, which suggest that this compound is not selective. Could NSGM 26 inhibit other MMPs?
  6. The authors could include in the discussion section the limitation of the study.

Reviewer 2 Report

I would like to congratulate the authors on their discovery that only two novel small molecule glycosaminoglycans (GAG) mimicking sulfated scaffolds, the sulfated benzofuran 26 and sulfated quinazolinone 38, were identified as potential synthetic inhibitors of matrix metalloproteinase -8 (MMP-8) with different levels of inhibition abilities.

MMPs play a significant role in physiological growth and development and in a number of diseases, e.g. atherosclerosis, tumor growth, cardiovascular disease, kidney diseases etc. Studies in vitro and in vivo suggest that MMPs are excellent targets for synthetic inhibitors. MMP inhibition can be overcome by enhanced MMP production at the transcriptional or translational levels, or that another MMP can intervene and cleave the components of the extracellular matrix when the specific MMP is inhibited for prolonged period.

Synthetic inhibitors generally contain a chelating group that binds the catalytic zinc atom at the MMP active site tightly. Other inhibitors are usually designed to interact with various binding pockets on the MMP of interest resulting in more or less specific inhibitory potentials for given MMPs.

It would be useful to enlarge the part devoted to the implication of MMP-8 with various diseases.

I suggest discussing the findings of MMP-8 in patients with coronary or carotid atherosclerosis as compared with individuals without evidence of atherosclerosis in the respective artery (Ye S. Putative targeting of matrix metalloproteinase-8 in atherosclerosis. Pharmacol Ther. 2015 Mar;147:111-122; Kato R, Momiyama Y, Ohmori R, Taniguchi H, Nakamura H, Ohsuzu F. Plasma matrix metalloproteinase-8 concentrations are associated with the presence and severity of coronary artery disease. Circ J. 2005 Sep;69(9):1035-40.) Look at the implication of MMP-8 in renal diseases, for instance urinary MMP-8 activity correlates with the severity of diabetic nephropathy, and so it might be a useful early marker of diabetic renal disease. (Nynke J van der Zijl 1, Roeland Hanemaaijer, Maarten E Tushuizen, Roger K Schindhelm, Jeannette Boerop, Cees Rustemeijer, Henk J Bilo, Jan H Verheijen, Michaela Diamant Urinary matrix metalloproteinase-8 and -9 activities in type 2 diabetic subjects: A marker of incipient diabetic nephropathy? Clin Biochem . 2010 May;43(7-8):635-9) or the involvement of MMP-8 with restoration of baseline kidney health after ischemic kidney injury ( Rajit K. Basu, Emily Donaworth, Brian Siroky, Prasad Devarajan & Hector R. Wong  Loss of matrix metalloproteinase-8 is associated with worsened recovery after ischemic kidney injury, 2015, Renal Failure, 37:3, 469-475) and some other findings in laboratory and clinical medicine.

In the abstract it is stated that sulphated quinazolines exhibited > 10-fold selectivity for MMP-8 over MMP-9. This statement was mentioned in the last paragraph of the Introduction in the summary of the study (line71) but not in the Results and Discussion part. Could you elaborate this finding un the Results part?

In the abstract the results of the discovery are not very clear it is better to state that sulphated benzofuran 26 and sulphated quinazolinone 38 were discovered as promising MMP-8 inhibitors with various inhibition efficacies.  

In introduction in the last paragraph (lines 68-75) the authors already summarize the results of the study instead of providing solely the aim of the study.

The Results and Discussion part is mingled: the introductory information is mixed with the results of the study. I suggest the part named Rationale Behind the Development of Glycosaminogycan Mimetics as MMP-8 Inhibitors (line 79) to move to the Introduction part. The part entitled Sulfated Quinazolinones Selectively Target MMP-8 is misleading because this part depicts the screening of NGSM15-42 at 100 μM against MMP-9, not MMP-8.

In the part entitled Computational Analysis of the Preferred Site of NGSMs Binding to MMP-8 (line 210)  I could not find the Figures referred (Figures 4A and 4B (line 216, line 218), they are not available in the main manuscript or the supplementary material.

Generally, it would be better if the authors better organise the framework of the manuscript into: Abstract, Introduction, Rationales of the Study. Materials and Methods. Results. Discussion, and Conclusion.

I suggest focusing firstly on the rationales of the study, secondly on the results of the study, then on discussion and finally on the conclusion of the study. For example, in the conclusion part only the first paragraph (lines 286-291) depicts the summary of the findings; the second paragraph (lines 292-307) is potential perspective/recommendation for future research and should be moved the Discussion part. The Conclusion should be concise and clear.

It would be better if the authors rewrite the manuscript in a concise manner with more emphasis on the significance of the discovery.  

Line 45 ADAM and ADAMTS abbreviations used with no explanation of the meaning

Table 1 COPD again abbreviation used with no explanation of the meaning

The titles of the subchapters should be uniform - sometimes you use the capitalized words but not in all main words (line79) or you do not use the capitalized main words (line 119)

I could not find Figures 4A and 4B.

There is no reference to Figure 5B in the main text (line239).

In the Experimental Section the web pages of the Companies where the reagents were purchased should be mentioned.

In Figure S1 do no use doesn’t but does not

I recommend thanking other personnel who participated in the study - is it true that the first author did all the laboratory, inhibition, and computational studies alone with no assistance from other colleagues and department personnel?

Reviewer 3 Report

Dear Authors,
the paper entitled "Discovery of Sulfated Small Molecule Inhibitors of Matrix Metalloproteinase-8" is well structured and methodologically supported. The results are well integrated into the discussion.

1. I would suggest enriching the introduction with some more information about the matrix metalloproteinases.

2. Why do you use a gelatinase like MMP-9 as a benchmark for selectivity? A comparison with MMP-1 or MMP-13 could not be more appropriate? It could be appropriate to explain the choice.
